# Hepatocyte-Specific *Phgdh*-Deficient Mice Culminate in Mild Obesity, Insulin Resistance, and Enhanced Vulnerability to Protein Starvation

**DOI:** 10.3390/nu13103468

**Published:** 2021-09-29

**Authors:** Momoko Hamano, Kayoko Esaki, Kazuki Moriyasu, Tokio Yasuda, Sinya Mohri, Kosuke Tashiro, Yoshio Hirabayashi, Shigeki Furuya

**Affiliations:** 1Department of Bioscience and Bioinformatics, Faculty of Computer Science and Systems Engineering, Kyushu Institute of Technology, Fukuoka 820-8502, Japan; 2Laboratory of Functional Genomics and Metabolism, Faculty of Agriculture, Kyushu University, Fukuoka 819-0395, Japan; 3Laboratory for Neural Cell Dynamics, RIKEN Center for Brain Science, Wako 351-0198, Japan; kayoko.esaki@riken.jp; 4Department of Bioscience and Biotechnology, Graduate School of Bioresource and Bioenvironmental Sciences, Kyushu University, Fukuoka 819-0395, Japan; momotarosan28@gmail.com (K.M.); yasuda.tokio.221@s.kyushu-u.ac.jp (T.Y.); mintonazure@gmail.com (S.M.); ktashiro@grt.kyushu-u.ac.jp (K.T.); 5Laboratory of Molecular Gene Technology, Faculty of Agriculture, Kyushu University, Fukuoka 819-0395, Japan; 6Innovative Bio-Architecture Center, Faculty of Agriculture, Kyushu University, Fukuoka 819-0395, Japan; 7Cellular Informatics Laboratory, RIKEN, Wako 351-0198, Japan; hirabaya@riken.jp; 8Institute for Environmental and Gender-Specific Medicine, Juntendo University Graduate School of Medicine, Chiba 279-0021, Japan

**Keywords:** *Phgdh*, liver, l-serine deficiency, insulin signaling, glucose tolerance

## Abstract

l-Serine (Ser) is synthesized de novo from 3-phosphoglycerate via the phosphorylated pathway committed by phosphoglycerate dehydrogenase (*Phgdh*). A previous study reported that feeding a protein-free diet increased the enzymatic activity of Phgdh in the liver and enhanced Ser synthesis in the rat liver. However, the nutritional and physiological functions of Ser synthesis in the liver remain unclear. To clarify the physiological significance of de novo Ser synthesis in the liver, we generated liver hepatocyte-specific *Phgdh* KO (LKO) mice using an albumin-Cre driver. The LKO mice exhibited a significant gain in body weight compared to Floxed controls at 23 weeks of age and impaired systemic glucose metabolism, which was accompanied by diminished insulin/IGF signaling. Although LKO mice had no apparent defects in steatosis, the molecular signatures of inflammation and stress responses were evident in the liver of LKO mice. Moreover, LKO mice were more vulnerable to protein starvation than the Floxed mice. These observations demonstrate that *Phgdh*-dependent de novo Ser synthesis in liver hepatocytes contributes to the maintenance of systemic glucose tolerance, suppression of inflammatory response, and resistance to protein starvation.

## 1. Introduction

l-Serine (Ser), a nutritionally dispensable amino acid, serves as an indispensable metabolite essential for mammalian fetal development [1,2]. Ser is synthesized de novo from 3-phosphoglycerate, which is catalyzed by the phosphorylated pathway composed of 3-phosphoglycerate dehydrogenase (*Phgdh*), phosphoserine aminotransferase 1 (*Psat1*), and phosphoserine phosphatase (*Psph*). Ser is utilized for the synthesis of important metabolic components, such as glycine, nucleotides, glutathione, tetrahydrofolate derivatives, and membrane lipids. We have previously demonstrated the physiological significance of de novo Ser synthesis at the cellular level. Extracellular Ser limitation leads to cell growth arrest and cell death, which is associated with the enhanced phosphorylation of p38MAPK and SAPK/JNK in *Phgdh*-deficient embryonic fibroblasts (KO-MEFs) under conditions of reduced intracellular Ser availability [3]. Simultaneously, p38MAPK is activated in part by 1-deoxy-sphinganine (doxSA), an atypical sphingolipid, which is synthesized by serine palmitoyltransferase [3,4]. Intracellular Ser deficiency induces the rapid activation of the integrated stress response (ISR) pathway, altered expression profiles of Atf4-dependent in KO-MEFs, and enhanced vulnerability to oxidative stress with the simultaneous upregulation of inflammatory gene expression [5]. Moreover, we demonstrated that extracellular Ser limitation led to a transient transcriptional activation of Atf4-target genes, including cation transport regulator-like protein 1 (*Chac1*) in a mouse hepatocarcinoma cell line expressing Phgdh [6]. These observations indicate that reduced availability of Ser by *Phgdh* disruption elicits a wide variety of stress and injury responses in non-malignant cells, at least in an in vitro cell culture setting. 

In parallel with these cell culture studies, our in vivo studies demonstrated that systemic *Phgdh* deletion resulted in severe intrauterine growth retardation and embryonic lethality, which recaptures the major symptoms of human Neu-Laxova syndrome patients. Unlike conventional KO mice, brain-specific *Phgdh* knockout (KO) mice were able to escape from embryonic lethality but exhibited marked reductions in both l- and D-Ser levels in the brain, which were accompanied by a diminished *N*-methyl-D-aspartate (NMDA) receptor function [7]. These observations reinforce the importance of de novo Ser synthesis in physiological processes, including embryonic development and mature brain functions. However, Phgdh is expressed at higher levels in certain tissues, including the heart, kidney, muscle, and liver [8]. Among these tissues, the expression and activity of the phosphorylated pathway in the liver are regulated by systemic levels of protein/amino acid nutrition and hormones [7,8]. Recent studies have implicated that Phgdh-dependent Ser synthesis supports general lipid homeostasis [9] and appears to prevent non-alcoholic fatty liver disease [10,11,12]. Nonetheless, it remains unclear how de novo Ser synthesis via the phosphorylated pathway contributes to the physiological function of the liver at steady state. 

To clarify the physiological significance of de novo Ser synthesis in the liver, we generated liver hepatocyte-specific *Phgdh* KO (LKO) mice using an albumin-Cre driver. In this study, we demonstrated that hepatocyte-specific *Phgdh*-deficient mice led to mild obesity, deteriorated glucose tolerance, and increased mortality when fed a protein-free diet.

## 2. Materials and Methods

### 2.1. Hepatocyte-Specific Phgdh Knockout Mice

Mice with homozygous conditional *Phgdh* alleles (*Phgdh Phgdh ^flox/flox^*), hereafter called Floxed, were obtained as previously described [7]. The presence of a conditional allele (*Phgdh^flox^*) was identified by PCR using tail DNA with a primer pair directed against the third intron (forward primer, 5′-CATGAGGAA CTGAACTGAAGGATTGA-3′; reverse primer, 5′-CAAGGAGGCTCACACATCCCAGAAC-3′), which generated a 310 bp amplicon, and the fourth and fifth intron (forward primer, 5′-CATGAGGAACTGAACTGAAGGATTGA-3′; reverse primer, 5′-CTTCAGCTCTCATGGCAGACGAGCA-3′), which generated a 350 bp amplicon. Mice conditionally lacking *Phgdh* in hepatocytes (*Albumin* (Alb)*^+/Cre^*;*Phgdh^flox/flox^*), hereafter called LKO, were obtained by interbreeding female Floxed mice with male Floxed mice carrying the Alb-Cre transgene, which were generated by crossing Floxed mice with Alb-Cre transgenic mice. To detect the Cre transgene used in this study, PCR was carried out with the primers 5′-AATTTGCCTGCATTACCGGTCGATGCAACG-3′ and 5′-CCATTTCCGGTTATTCAACTTGCACCATGC-3′, which generated a 190 bp amplicon of part of the Cre-coding region. Unlike conventional *Phgdh* knockout mice (*Phgdh^-/-^*), LKO pups were born at the expected Mendelian ratio when female Floxed mice were crossed with male LKO mice (data not shown). Littermates with the Floxed genotype (*Phgdh ^flox/flox^*) were used as controls. 

LKO and Floxed mice were maintained in a 12-h light/dark cycle with unlimited access to water and laboratory chow containing 20% casein. The animal experimental protocols for this study were approved by the Animal Ethics Committees of the RIKEN Center for Brain Science (H25-2-241) and Kyushu University (A27-103).

### 2.2. Glucose Tolerance Test

Glucose solution was prepared as 2.5 g/10 mL in saline and sterilized by filtration. A glucose tolerance test was performed by intraperitoneally injecting glucose (2 g/kg body weight) into mice after overnight fasting. Tail-vein blood samples were collected at 0 (prior to administration) and 30, 60, 90, and 120 min after glucose administration. Blood glucose levels were measured using an ACCU-CHEK system (Roche Diagnostics, Tokyo, Japan).

### 2.3. Feeding of Protein-Free Diet

Mice were maintained in a 12-h light/dark cycle with unlimited access to food and water. Male Floxed and LKO mice (10 weeks old, *n* = 5 each) were fed a protein-free diet (TestDiet, #5765; casein-vitamin free 0%, sucrose 36.15%) and normal diet (TestDiet, #5755; Casein-vitamin free 21%, Sucrose 15%) for 3 weeks, and their body weights were measured daily. 

### 2.4. RNA Isolation and Microarray Analysis

Total RNA was extracted from the livers of LKO and littermate Floxed mice (male, 30 weeks old, *n* = 6, each genotype) using the RiboPure^TM^ Kit (Thermo Fisher Scientific, Tokyo, Japan) according to the manufacturer’s instructions. After the extraction of total RNA, the concentration of RNA was measured using a NanoDrop LITE spectrophotometer (Thermo Fisher Scientific). Total RNA (5 µg) was treated with DNase (TURBO DNA-free^TM^, Thermo Fisher Scientific). After DNase treatment, the quality of treated RNA was assessed using an RNA Nano Chip and a 2100 Bioanalyzer (Agilent Technologies, Santa Clara, CA, USA). Then, 200 ng of total RNA was reverse-transcribed into double-stranded cDNA, then transcribed and labeled with Cyanine 3-CTP using T7 RNA Polymerase. Next, 1.65 mg of the purified cyanine 3-labeled cRNA was hybridized to mouse GE 4 × 44 K v2 Microarrays (Agilent Technologies, Tokyo, Japan) according to the manufacturer’s protocol. The signal intensity was measured using a G2565CA Microarray Scanner System (Agilent Technologies). The processed intensities were normalized across the samples and loaded using quantile normalization. All microarray data were submitted to the Gene Expression Omnibus (accession number GSE179912). The raw signal intensities of all samples were log_2_-transformed and normalized by quantile algorithm with ‘preprocessCore’ library package [13] on Bioconductor software. We selected the probes, excluding the control probes, where the detection *p*-values of all samples were less than 0.01 and used them to identify differentially expressed genes. We then applied the Linear Models for Microarray Analysis (limma) package [14] of Bioconductor software. The criteria were limma *p* < 0.05, between LKO and Flox liver samples.

### 2.5. KEGG Pathway Enrichment Analysis for Differentially Expressed Genes

The Database for Annotation, Visualization and Integrated Discovery (DAVID) (https://david.ncifcrf.gov/ (accessed on 14 March 2020)) [15] was used for KEGG pathway enrichment analysis of differentially expressed genes (DEGs) in LKO mice whose expression was upregulated or downregulated compared to Floxed mice. The top GO terms and KEGG pathway in the annotation clusters that ranked in the functional annotation clustering function with statistical significance (*p* < 0.05) were extracted. The enrichment *p*-values of all extracted GO terms and KEGG pathways for each module were calculated using DAVID.

### 2.6. Gene Ontology Enrichment Analysis for Differentially Expressed Genes

Gene set enrichment analysis (GSEA) was used to determine whether a priori defined sets of genes showed significantly enriched GO terms. GSEA was also used to identify the GO terms associated with significantly enriched upregulated or downregulated genes in the liver of LKO mice.

### 2.7. Ingenuity Pathways Analysis

Biologically relevant networks were created using the Ingenuity Pathways Analysis (IPA) program (http://www.Ingenuity.com (accessed on 15 April 2015)) [16] as previously described. Based on algorithmically generated connectivity between gene–gene, gene–protein, and protein–protein interactions, the program develops functional molecular networks that overlay genes in the dataset. This program calculated *p*-values for each network by comparing the number of focus genes that were mapped in a given network relative to the total number of occurrences of those genes in all networks. The score for each network is shown as the negative log of the *p*-value, which indicates the likelihood of finding a set of genes in the network by random chance.

### 2.8. Quantitative Analysis of mRNA Expression

Total RNA was extracted from the liver at 30 weeks using the Ribo Pure kit (Thermo Fisher Scientific, Waltham, MA, USA), as described above. Following isolation, 1 μg of DNase-treated RNA was used to generate cDNA by reverse transcription using the High-Capacity cDNA Reverse Transcription kit (Applied Biosystems, Life Technologies Japan Ltd., Tokyo, Japan). Quantitative real-time PCR was performed using a Model Mx3000P Real Time PCR system (Agilent Technologies Japan Ltd, Tokyo, Japan) containing Thunderbird SYBR qPCR Mix (TOYOBO, Osaka, Japan) and the reference dye ROX according to the manufacturer’s recommendations. Primer sequences used were *Phgdh* (forward, 5′-TGGAGGAGATCTGGCCTCTC-3′, and reverse, 5′-GCCTCCTCGAGCACAGTTCA-3′), 6-phosphofructo-2-kinase/fructose-2,6-biphosphatase 3 (*Pfkfb3*) (forward, 5′-CTACGAGCAGTGGAAGGCACTC-3′, and reverse, 5′-AATTCCATGATCACAGGCTCCA-3′), pyruvate dehydrogenase lipoamide kinase isozyme 4 (*Pdk4*) (forward, 5’-ACCGCATTTCTACTCGGATGC-3′, and reverse, 5′-CGCAGAGCATCTTTGCACACT-3′), early growth response protein 1 (*Egr1*) (forward, 5′-CCGAGCGAACAACCCTATGA-3’, and reverse, 5′-GTCATGCTCACGAGGCCACT-3′), and fatty acid synthase (*Fasn*) (forward, 5′-TTCCAAGACGAAAATGATGC-3′, and reverse, 5′-AATTGTGGGATCAGGAGAGC-3′). All reactions were performed in triplicate. Data analysis was carried out using the cycle threshold values of target gene expression normalized to actin as the internal control.

### 2.9. Western Blot Analysis

Samples of liver at 30 weeks were homogenized in a buffer containing 1.25 mM Tris-HCl (pH 7.6), 150 mM NaCl, 1% NP40, 1% sodium deoxycholate, 0.1% SDS, a protease inhibitor cocktail (Nacalai Tesque), and a phosphatase inhibitor cocktail (Nacalai Tesque). Homogenates were centrifuged at 20,000× *g* for 10 min to obtain total protein extracts, and concentrations were determined using a Protein Assay Bicinchoninate kit (Nacalai Tesque). Protein samples were fractionated by 7.5% SDS-polyacrylamide gel electrophoresis and transferred onto a polyvinylidene fluoride membrane (Bio-Rad, Hercules, CA, USA). Blotted proteins were probed with the following primary antibodies: anti-Phgdh (rabbit, 0.3 g/mL, provided by Dr. M. Watanabe at Hokkaido University), which is cross-reactive with the mouse homolog, anti-NF-κB (also known as Nuclear factor kappa-light-chain-enhancer of activated B cells) (rabbit monoclonal, 1:1000; Cell Signaling Technology, Danvers, MA, USA), anti-phospho NF-κB Ser 536 (rabbit monoclonal, 1:1000; Cell Signaling Technology, Danvers, MA, USA), anti-Akt (also known as protein kinase B) (rabbit polyclonal, 1:1000; Cell Signaling Technology, Danvers, MA, USA), anti-phospho Akt Thr 308 (rabbit polyclonal, 1:1000; Cell Signaling Technology, Danvers, MA, USA), anti-GSK3β (also known as glycogen synthase kinase 3β) (rabbit monoclonal, 1:1000; Cell Signaling Technology, Danvers, MA, USA), anti-phospho GSK3β Ser 9 (rabbit polyclonal, 1:1000; Cell Signaling Technology, Danvers, MA, USA), anti-IRS1 (also known as insulin receptor substrate 1) (rabbit monoclonal, 1:1000; Cell Signaling Technology, Danvers, MA, USA), anti-phospho IRS1 Ser 612 (rabbit monoclonal, 1:1000; Cell Signaling Technology, Danvers, MA, USA), anti-phospho IRS1 Ser 636/639 (mouse 632/635) (rabbit polyclonal, 1:1000; Cell Signaling Technology, Danvers, MA, USA), anti-Erk1/2 (also known as extracellular signal-regulated kinase 1/2 and p44/42 MAPK) (rabbit monoclonal, 1:1000; Cell Signaling Technology, Danvers, MA, USA), anti-phospho Erk1/2 Thr 202/Thr 204 (rabbit monoclonal, 1:1000; Cell Signaling Technology, Danvers, MA, USA), anti-SAPK/JNK (also known as stress-activated protein kinase (SAPK)/jun amino terminal kinase (JNK)) (rabbit monoclonal, 1:250; Cell Signaling Technology, Danvers, MA, USA), anti-phospho SAPK/JNK (rabbit monoclonal, 1:250; Cell Signaling Technology, Danvers, MA, USA), anti-Egr1 (also known as early growth response protein 1) (rabbit monoclonal, 1:200; Cell Signaling Technology, Danvers, MA, USA), anti- β-Actin (mouse monoclonal, 1:500; FUJIFILM Wako Pure Chemical Corporation, Japan), and anti-Gapdh (mouse monoclonal, Chemicon, 1:50,000; Merck Millipore, Billerica, MA, USA). The bound antibodies were visualized with the Pierce SuperSignal West Pico Chemiluminescence Detection System (SuperSignal; Thermo Fisher Scientific) after incubation with the appropriate secondary antibodies conjugated with horseradish peroxidase (Cell Signaling Technology Japan K.K., Tokyo, Japan). The chemiluminescent signal was detected by exposure to X-ray films (FUJIFILM, Tokyo, Japan), and signal intensities were quantified using the CS Image Analyzer 3 software (ATTO Corp., Tokyo, Japan).

### 2.10. Histological Evaluation

Mice (male, 28–32 weeks old: *n* = 5 in each genotype) were anesthetized with isoflurane and perfused with 4% paraformaldehyde in 0.1 M sodium phosphate buffer (pH 7.2) after removing blood with 0.1 M sodium phosphate buffer (pH 7.2). The livers were post-fixed overnight in the same fixative. Hematoxylin and eosin staining was performed by Soshiki Kagaku, Lad, Inc. (Yokohama, Japan).

### 2.11. Serum Biochemical Test

Blood was collected after decapitation under anesthesia with isoflurane. Serum was prepared by centrifuging blood at 3000× *g* for 15 min after standing still for 2 h. Sample analysis of serum (male mice, 28–32 weeks old) was performed by the Health Sciences Research Institute East Japan Co., Ltd. (Saitama, Japan). The serum concentrations of lipoprotein (Lipopro), total cholesterol (Total-Cho), LDL cholesterol (LDL-C), HDL cholesterol (HDL-C), aspartate transaminase (AST), alanine transaminase (ALT), and non-esterified fatty acid (NEFA) were measured.

### 2.12. Amino Acid Analysis

The liver and kidney at 30 weeks were homogenized in 5 volumes of ultrapure water and centrifuged at 15,000 rpm for 30 min at 4 °C. The supernatant was mixed with a 1/10 volume of 5% perchloric acid (HClO_4_). The mixture was incubated on ice for 25 min and then centrifuged for 30 min at 15,000 rpm at 4 °C. The supernatant was neutralized by adding 1/10 volume of 8 N KOH. The sera were diluted 3-fold in sterilized water. Proteins in the supernatant and diluted sera were removed by adding a 1/10 volume of 60% perchloric acid, and the solution was adjusted to pH 7–8 with 8 M KOH. The amino acid composition of the supernatant was determined using an Acquity UPLC H-class system (Waters, Milford, MA, USA).

### 2.13. Statistical Analysis

To detect DEGs in the liver of LKO mice compared to Floxed mice, the fold change of gene expression was calculated compared to Floxed mice, using the thresholds of |log_2_Fold change| ≥ 1. To visualize the distributions of gene expression levels, a volcano plot was generated in R. Unpaired two-tailed Student’s *t*-test was applied using adjusted *p* < 0.05. Differences between two groups were examined using Student’s *t*-test were considered statistically significant at *p* < 0.05. All statistical analyses were performed using KaleidaGraph 4.0 (Synergy Software, Tokyo, Japan).

## 3. Results

To inactivate *Phgdh* in a liver-hepatocyte-specific manner, the *Phgdh* allele was disrupted by crossing the female Floxed mice (*Phgdh ^flox/flox^*) with male mice expressing Cre under the control of the *albumin* promoter (referred to as Alb-Cre mice). We obtained mice in subsequent generations with the Alb*^+/Cre^*;*Phgdh ^flox/flox^* genotype, referred to as LKO. The Floxed allele of *Phgdh* without exon deletion (*Phgdh ^flox^*) was detected in the livers of both Floxed and LKO mice, while the null allele lacking the fourth and fifth exons of *Phgdh* (*Phgdh*^-^) was detected only in the liver (Figure 1A), but not in the kidney of LKO mice (data not shown). The efficacy of Cre-mediated deletion of *Phgdh* was assessed by qRT-PCR and Western blotting. The mRNA and protein levels of *Phgdh* in the liver of LKO mice were lower than those in Floxed mice (Figure 1B,C). The expression of *Phgdh* mRNA and protein was significantly reduced but not completely abolished in the liver of LKO mice because the Cre-mediated recombination by the Alb-Cre transgene occurred only in hepatocytes, and the *Phgdh* expression was maintained in other cell types in the liver of LKO mice. 

To evaluate the in vivo phenotypes caused by *Phgdh* deletion in hepatocytes, we compared the body and organ weights of LKO and Floxed mice. Interestingly, LKO mice exhibited a subtle but significant weight gain compared to the Floxed mice after 23 weeks of age (Figure 2A). Body weight, organ weight of liver, epididymal white adipose tissue (eWAT), parametrial white adipose tissue (pWAT), and mesenteric white adipose tissue (mWAT) were significantly higher than those of Floxed mice at 30 weeks of age (Figure 2B and Appendix A). However, steatosis was not observed in the liver of LKO mice (Figure 2C,D). To clarify the alteration of liver function in LKO mice, we performed a biochemical analysis of liver markers in the serum. There were no significant changes in the concentrations of cholesterol, non-esterified fatty acid (NEFA), aspartate aminotransferase (AST), and alanine aminotransferase (ALT) in LKO mice (Figure 3A). To evaluate the effect on glucose metabolism, we performed a glucose tolerance test. The blood glucose levels in the LKO mice were significantly higher than those in the Floxed mice after intraperitoneal glucose injection (Figure 3B). Thus, hepatocyte-specific *Phgdh* deletion impaired glucose clearance and led to mild obesity without affecting the levels of liver biochemical markers in the serum.

We then examined the alterations in the amino acid metabolism in LKO mice. To our surprise, the Ser concentration in the liver of LKO mice was not altered compared to that of Floxed mice (Table 1A), while Ser concentration in LKO kidneys was significantly increased compared to Floxed mice (Table 1B). In addition, the concentrations of l-aspartic acid (l-Asp), l-histidine (l-His), l-arginine (l-Arg), l-alanine (l-Ala), l-tyrosine (l-Tyr), l-methionine (l-Met), l-phenylalanine (l-Phe), l-isoleucine, γ-aminobutyric acid, and l-leucine were increased in the kidneys of LKO mice (Table 1B). This suggests that amino acid metabolism in LKO mice was altered in the kidney and Ser availability in the liver of LKO mice was maintained via the supply of non-hepatic cells in the liver and/or kidney. 

To assess whether hepatocyte-specific *Phgdh* deletion in the liver modulates gene expression, we performed a microarray analysis of liver mRNA. We identified 2770 genes that were significantly differentially expressed in the liver of LKO mice compared to that of Floxed mice (see Materials and Methods). Among them, 1191 genes were downregulated (<0.90–0.17-fold) and 1579 genes were upregulated (>1.1–4.9-fold) (Appendix A). A KEGG pathway analysis by DAVID demonstrated that the PPAR signaling pathway, which regulates lipid metabolism in the liver, was enriched in the upregulated DEGs in the liver of LKO mice (Table 2A), while the phosphoinositide 3-kinase (PI3K)‒Akt signaling pathway, which is involved in intracellular insulin signaling, was enriched in the downregulated DEGs of the liver in LKO mice (Table 2B). An ingenuity pathway analysis (IPA) was also performed to identify the affected gene networks and canonical signaling pathways. By analyzing DEGs in the liver of LKO mice, IPA identified the IL-10 signaling network (Appendix A) and generated lists of significantly affected “canonical pathway” (Table 3A), “disease and disorder” (Table 3B), and “hepatotoxicity” (Table 3C). These include IL-12 signaling pathways in the top canonical pathway, multiple hepatotoxicity, and hepatic system diseases. These results suggest that *Phgdh* deletion in hepatocytes induces the dysregulation of multiple canonical pathways in the liver. Since it has been well documented that inflammation deteriorates systemic insulin sensitivity and obesity [17], we tested whether an inflammation-related response occurred in the liver of LKO mice. First, we examined the phosphorylation of nuclear factor-kappa B (NF-κB), which serves as an essential transcription factor for inflammatory responses. The phosphorylated NF-κB at Ser-536 showed an increasing trend in the liver of LKO mice compared to Floxed mice (Figure 4). These observations reveal the occurrence of inflammation-related molecular changes in the liver of LKO mice. 

A gene set enrichment analysis (GSEA) was used to identify functionally related groups of gene sets, annotated use systems, the Gene Ontology (GO) biological processes, and KEGG pathways in the liver of LKO mice. GSEA identified positively and negatively correlated gene sets in GO biological processes (GOBP) (Table 4A,B) and KEGG pathways (Table 4B,D). Several gene sets with higher normalized enrichment scores in positively correlated gene sets in GOBP “Negative regulation of nucleocytoplasmic transport,” “Fatty acid beta oxidation,” and “Electron transport chain” were considered as liver-relevant positively correlated gene sets (Table 4A and Figure 5A,B). In contrast, “Regulation of cytoplasmic translation,” “RNA phosphodiester bond hydrolysis exonucleolytic,” and “processes of RNA metabolism “were detected with higher normalized enrichment scores in negatively correlated gene sets of GOBP in the liver of LKO mice (Table 4B and Figure 5C,D). In addition, “Glutathione metabolism” and “Fatty acid metabolism” were considered as liver-relevant positively correlated KEGG pathways with higher normalized enrichment scores in the liver of LKO mice (Table 4C and Figure 5E,F). “RNA degradation” and “Basal transcription factors” were detected as negatively correlated KEGG pathways with higher normalized enrichment scores (Table 4D and Figure 5G,H). These results suggest that multiple biological processes, such as RNA metabolism, mitochondrial function, and energy metabolism, were altered by the hepatocyte-specific deletion of *Phgdh* in the liver.

Since the current KEGG pathway analysis in DAVID points to an alteration in the intracellular insulin signaling cascade in the liver of LKO mice (Table 2B), we examined protein phosphorylation of components in the cascade. Western blot analysis demonstrated a trend toward decreasing phosphorylation of Akt at Thr-308 (Figure 6A), and a significant reduction in the phosphorylation of GSK3β at Ser-9 (Figure 6B) in the liver of LKO mice. Glycogen synthase is negatively regulated by inhibitory phosphorylation by GSK3β, which is also negatively regulated by Akt phosphorylation at serine-9 [18,19]. Decreased phosphorylation levels of Akt and GSK3β coincide with impaired glucose tolerance in the liver of LKO mice. We then examined the phosphorylation status of insulin receptor substrate 1 (IRS-1) because serine/threonine phosphorylation of IRS-1 is closely related to insulin resistance [20,21,22]. Among them, it is well documented that the phosphorylation of IRS-1 at Ser-612 and Ser-632/635 residues is negatively correlated with insulin signaling [23,24]. The phosphorylation levels of Ser-612 and Ser-632/635 were significantly higher in the livers of LKO mice than in Floxed mice (Figure 6C,D). Taken together, these observations indicate insulin resistance in the liver of LKO mice.

To examine the dysregulation of kinases directing IRS-1, we measured the phosphorylation of Erk1/2 and stress-activated protein kinase (SAPK)/Jun amino-terminal kinase (JNK), which phosphorylates the Ser-632/635 residues of IRS1 [20]. The phosphorylation of Erk1/2 and SAPK/JNK was significantly increased in the liver of LKO mice compared to Floxed mice (Figure 7A,B). Since the phosphorylation of Erk1/2 is regulated by early growth response (Egr)-1 in type 2 diabetic mice [25], we compared the Egr-1 expression in the liver of LKO and Floxed mice. The Egr-1 mRNA and protein levels showed a significant increase (Figure 7C) and a trend toward increasing levels, respectively, in the liver of LKO mice (Figure 7D). These results suggest that the activation of Egr-1 the Erk1/2 axis contributes to the negative regulation of insulin signaling at IRS-1 in the liver of LKO mice.

To gain an insight into the dysregulation of the enzymes regulating glucose metabolism under diminished insulin signaling in the liver of LKO mice, we examined the mRNA expression of 6-phosphofructo-2-kinase/fructose-2,6-biphosphatase 3 (*Pfkfb3*) and pyruvate dehydrogenase kinase 4 (*Pdk4*) as insulin-regulated regulators of glycolysis and gluconeogenesis, respectively [26,27]. *Pfkfb3* encodes 6-phosphofructo-2-kinase/fructose-2,6-bisphosphatase synthesizing fructose 2,6-bisphosphate, the glycolytic activator that promotes the conversion of fructose 1,6-bisphosphate to fructose 6-phosphate, a rate-limiting reaction in the glycolytic pathway [28]. Pdk4 inhibits the conversion of pyruvate to acetyl-CoA by inhibiting the phosphorylation of pyruvate dehydrogenase, which leads to suppression of glucose oxidation in the glycolytic pathway [29]. The mRNA levels of *Pfkfb3* in the liver were significantly reduced in LKO mice compared to Floxed mice (Figure 8A), while the mRNA levels of *Pdk4* in the liver were significantly increased in LKO mice compared to Floxed mice (Figure 8B). These gene expression profiles suggest diminished glucose oxidation and coincide with diminished insulin signaling in the livers of LKO mice. In addition to these results, we evaluated fatty acid synthesis to measure Fasn mRNA levels in the liver and adipose tissue. The mRNA levels of *Fasn* were not increased in the liver (Figure 8C) but increased in the eWAT (Figure 8D). These results suggested that steatosis did not progress, but fat accumulation in the adipose tissue was enhanced.

We then evaluated the alteration in tolerance to dietary protein starvation in LKO mice. LKO mice were fed a protein-free diet for 21 days, and their body weight (Figure 9A) and survival rate (Figure 9B) were found to change as a result. LKO mice showed a significantly greater weight loss than Floxed mice throughout the feeding of the protein-free diet (Figure 9A). The survival rate after feeding protein-free diet for 21 days was markedly lower in LKO mice than in Floxed mice (Figure 9B). LKO mice exhibited diarrhea and swelling of the small intestine 21 days after feeding the protein-free diet (data not shown). These observations suggest that the loss of *Phgdh* in hepatocytes leads to increased vulnerability to short-term protein malnutrition. 

## 4. Discussion

Recent studies in mice and humans suggest that the altered expression of enzymes composed of the phosphorylated pathway of de novo Ser synthesis in the liver is associated with fatty liver disease. Bioinformatic approaches have indicated the downregulation of PHGDH in the liver of patients with non-alcoholic fatty liver disease [10] and alcoholic hepatitis patients [30]. Diet-induced fatty liver model mice also exhibited a reduced expression of Phgdh in the liver [12]. Interestingly, the fatty acid treatment at high concentrations (500–700 μM) caused a downregulation of Phgdh in isolated hepatocytes [12]. Although these observations raise the possibility that the downregulation of hepatic Phgdh may be implicated in the onset and/or progression of fatty liver disease, the pathophysiological consequences of genetic *Phgdh* disruption in liver hepatocytes have not been previously explored experimentally. The present study demonstrates for the first time that hepatocyte-specific *Phgdh* deletion resulted in body weight gain and increased adipose tissue weights in mice at 23–30 weeks of age but did not cause the ectopic accumulation of fatty acids in the liver and/or increases in blood fatty acids and other lipids. These phenotypes implicate a regulatory role for Phgdh expressed in hepatocytes in systemic glucose metabolism, whereas the onset and development of fatty acid deposition in the liver are not directly promoted by downregulation of *Phgdh* in hepatocytes. The present observations strongly suggest that reduced Phgdh expression in hepatocytes alone is not sufficient to induce ectopic fat accumulation in the liver. Presumably, since the liver is composed of several different types of cell populations in addition to hepatocytes, the downregulation of Phgdh in other liver cell types may also be necessary to induce fat accumulation. To evaluate the pathobiological role of Phgdh in fatty liver, it will be necessary to delete or downregulate *Phgdh* in all cell types of the liver and then to identify the cell types involved in using liver cell type-specific mutants.

Clinical and experimental studies have shown that hepatic insulin resistance is strongly associated with non-alcoholic fatty liver disease (NAFLD) [31,32]. It was apparent that the insulin signal cascade was downregulated in the liver of LKO mice (Figure 10), as indicated by the reduced phosphorylation levels of Akt and GSK3β with enhanced phosphorylation of Ser residues in IRS1 (Figure 5) and Pfkfb3 downregulation (Figure 8A). However, ectopic lipid accumulation did not occur in the liver of LKO mice. Accumulating evidence has established a causal role for inflammation in the development of insulin resistance in insulin-responsive tissues, including the liver [33]. A bioinformatics analysis of gene expression profiles by IPA showed that an increased production and signaling of the pro-inflammatory cytokine interleukin (IL)-12 in macrophages was the top canonical pathway of upregulated genes in the liver of LKO mice (Table 3A). The IL-12 family of cytokines is known to be a potential inflammatory mediator linking obesity and insulin resistance. IL-12 family molecules and their receptors have been reported to be upregulated in insulin-responsive tissues, including the liver, under obese conditions in experiments using the Ob/Ob model [34]. The increased phosphorylation of NF-κB was also indicative of inflammation-like changes in the liver of LKO mice (Figure 4). Regarding the link between changes in Ser availability and NF-κB activation, Wang et al. reported that the NF-κB phosphorylation levels were increased in the brains of mice fed a Ser/Gly restricted diet (SGRD) and further increased when D-glucose was administered in combination with SGRD to induce inflammation through the intestinal tract [35]. In addition, macrophages play a multifaceted role in the regulation of inflammatory responses, and it was reported that extracellular Ser restriction leading to a marked reduction of Ser within the cells increases the expression of pro-inflammatory cytokines and suppresses the expression of IL-10, which exerts anti-inflammatory effects [36]; however, other studies observed a seemingly contradictory phenomenon in which the expression of the pro-inflammatory cytokine IL-1β depends on Ser metabolism in macrophages [37,38]. Erg1, an immediate early gene encoding zinc-finger transcription factor, responds quickly to a variety of stimuli, such as injury, growth factors, cytokines, and physical insults, and participates in tissue injury and repair in the liver via its downstream target genes [39]. Indeed, Egr-1 was reported to induce liver inflammation in a cholestatic injury model [40]. Hence, the induction of Egr1 mRNA and protein together with the inflammatory gene signature and enhanced NF-κB phosphorylation supports the notion that inflammation-like changes were elicited in the liver of LKO mice.

Although the cause of inducing inflammatory-like changes remains unknown, the GSEA analysis of the genes upregulated in the liver of LKO mice indicated significant changes in the gene sets of the glutathione metabolism and the mitochondrial electron transport chain system. Previously, we reported that extracellular Ser restriction decreased total glutathione content and led to the generation of H_2_O_2_ and induction of inflammation-related gene expression in mouse embryonic fibroblasts deficient in *Phgdh* [5]. Recent studies also demonstrated that *Phgdh* deletion elicited an inflammation-like response through the production of cytokines via an increase in reactive oxygen species in chondrocytes [41], while orally administered Ser was able to protect LPS-induced intestinal inflammation via p53-mediated glutathione synthesis [42]. Hence, it is presumed that a decrease in the glutathione content and/or a dysregulation of the mitochondrial electron transport system lead to oxidative stress in the liver of LKO mice, which may trigger inflammation-like responses in the liver of LKO mice. In particular, Kupffer cells are likely to be candidate cells mediating these responses because they have macrophage-like functions in the liver. Further studies are needed to investigate whether the genetic inactivation of Phgdh elicits oxidative stress and/or inflammatory responses in the different cell types of the liver.

It should be noted that the results of the amino acid analysis performed in this study were unexpected, and no significant differences in Ser content in the liver or free Ser concentration in the serum were observed in LKO mice compared to those in Floxed control mice. On the other hand, the Ser content in the kidney was increased by 1.4-fold compared to that in Floxed mice (Table 1B), and the Phgdh protein levels were also increased significantly by 1.15-fold (data not shown). In addition, Ser levels in the soleus muscle was increasing (Mohri, Hamano, Furuya, data not shown). These changes may reflect the upregulation of de novo Ser synthesis in the kidney and muscle via the phosphorylated pathway to supply the liver. Similarly, cell types other than hepatocytes may also augment de novo Ser synthesis in the liver of LKO mice. A similar phenomenon has been observed in organs other than the liver. Mutations in PHGDH have been identified as the cause of a rare neurodegenerative degenerative retinal disease, macular telangiectasia type 2 (MacTel), which results in photoreceptor degeneration in humans [43]. Shen et al. recently reported that the Müller cell-specific deletion of the *Phgdh* gene in mice caused photoreceptor degeneration but did not cause a reduction in the retinal Ser levels, rather, it markedly increased amino acid levels [44]. This may be due to the enhanced expression of Phgdh in microglia and macrophages in the retina, which were recruited and activated by photoreceptor degeneration, and most likely compensated for Ser in the retina. Likewise, the hepatocyte-specific inactivation of Phgdh was also likely to elicit a compensatory supply of Ser within the liver, kidney, muscle, and other organs. There is a possibility that this compensatory Ser synthesis arises from cells mediating inflammatory responses in the liver of LKO mice in the same way as microglia and microphages in the retina with Müller cell-specific deletion of Phgdh. 

This study demonstrated that LKO mice exhibited increased mortality compared to Floxed mice on a protein-free diet (Figure 8B), which was accompanied by diarrhea (data not shown). These consequences of LKO mice fed the protein-free diet strongly suggest that de novo Ser synthesis via Phgdh in hepatocytes plays a crucial role for adapting to a protein starved nutritional condition. Furthermore, the present observations raise the possibility that Phgdh in hepatocytes preserves intestinal integrity under protein-starved nutritional conditions via an unknown mechanism. It was demonstrated that the enzymatic activity and mRNA of Phgdh increased quickly in rats fed a protein-free diet [45,46], although the functional significance of Phgdh induction by a protein-free diet has remained unclear for a long time. The intake of a protein-free diet was shown to impair the morphological architecture and function of the intestine [47], while a recent study observed that oral supplementation of Ser maintained morphological and functional integrity of the intestine and prevented diarrhea incidence in piglets [42]. Given these previous observations, together with the present findings, the intake of a protein-free diet appears to be rapidly sensed by hepatocytes. Immediately thereafter, intestinal function is maintained via an unknown liver–small intestine correlation mechanism mediated by Ser. More detailed experimental studies are needed to understand the molecular basis of this phenomenon.

In conclusion, this study reveals that de novo serine synthesis initiated by Phgdh in liver hepatocytes contributes to maintaining the insulin/IGF signaling pathway in the liver, systemic glucose tolerance, and resilience to protein deprivation. Since an impairment in systemic glucose tolerance serves as a high risk for developing type 2 diabetes mellitus in humans, it is anticipated that the elucidation of the pathobiological relationship between the decrease in serine synthesis via the phosphorylation pathway, the diminishment of the insulin/IGF signaling pathway, and the impaired glucose tolerance in the human liver will provide new mechanisms related to impaired glucose tolerance and useful opportunities for its diagnosis and prevention.

## Figures and Tables

**Figure 1 nutrients-13-03468-f001:**
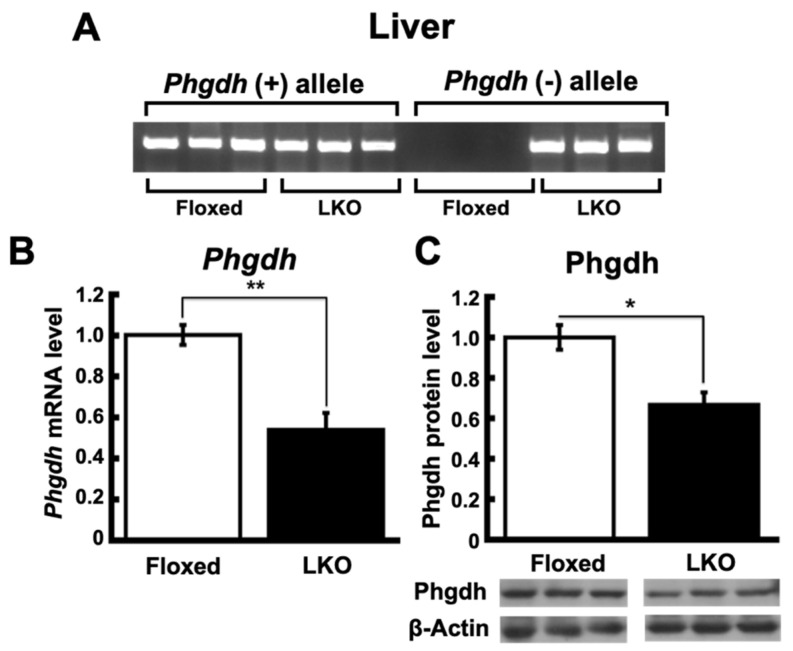
Targeted inactivation of *Phgdh* in liver. (**A**) Genotyping PCR from *Phgdh*(+) allele and *Phgdh*(−) allele in the liver of Floxed and LKO mice at 30 weeks (*n* = 6 each). (**B**) mRNA level of Phgdh in the liver of Floxed (*n* = 3) and LKO (*n* = 4) mice at 30 weeks. (**C**) Protein level of Phgdh in the liver of Floxed (*n* = 4) and LKO (*n* = 6) mice at 30 weeks. Comparable staining of β-actin was used to verify equivalent protein loading. Student’s *t*-test, * *p* < 0.05, ** *p* < 0.005.

**Figure 2 nutrients-13-03468-f002:**
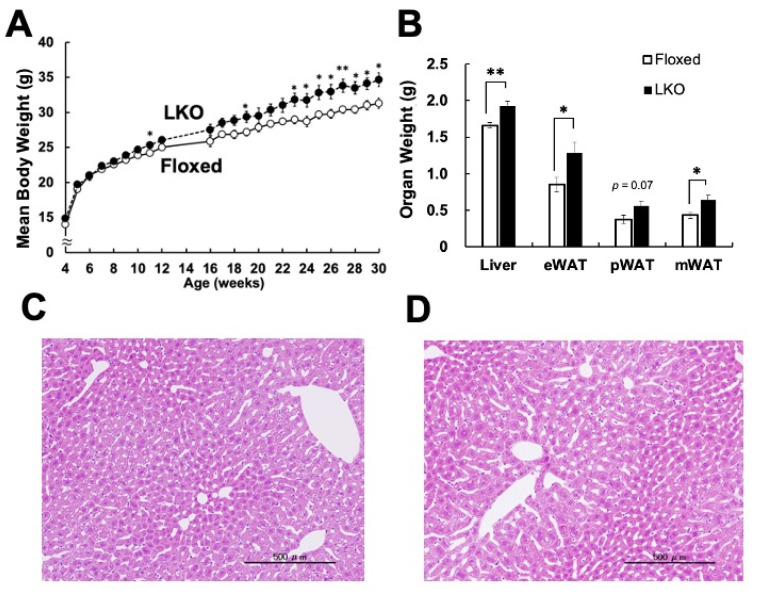
Hepatocyte-specific deletion of Phgdh causes the weight gain. (**A**) Growth curve of Floxed (white node) and LKO (black node) mice from 4 to 30 weeks (*n* = 6 each). (**B**) The body weight and the organ weight of the liver, epididymal white adipose tissue (eWAT), parametrial white adipose tissue (pWAT), and mesenteric white adipose tissue (mWAT) were measured in Floxed and LKO mice at 30 weeks (*n* = 6 each). (**C**,**D**) Histological evaluation of the liver in Floxed (**C**) and LKO (**D**) mice at 28 weeks (*n* = 5 each). Representative images are shown. Scale bar, 500 μm. Student’s *t*-test, * *p* < 0.05, ** *p* < 0.005.

**Figure 3 nutrients-13-03468-f003:**
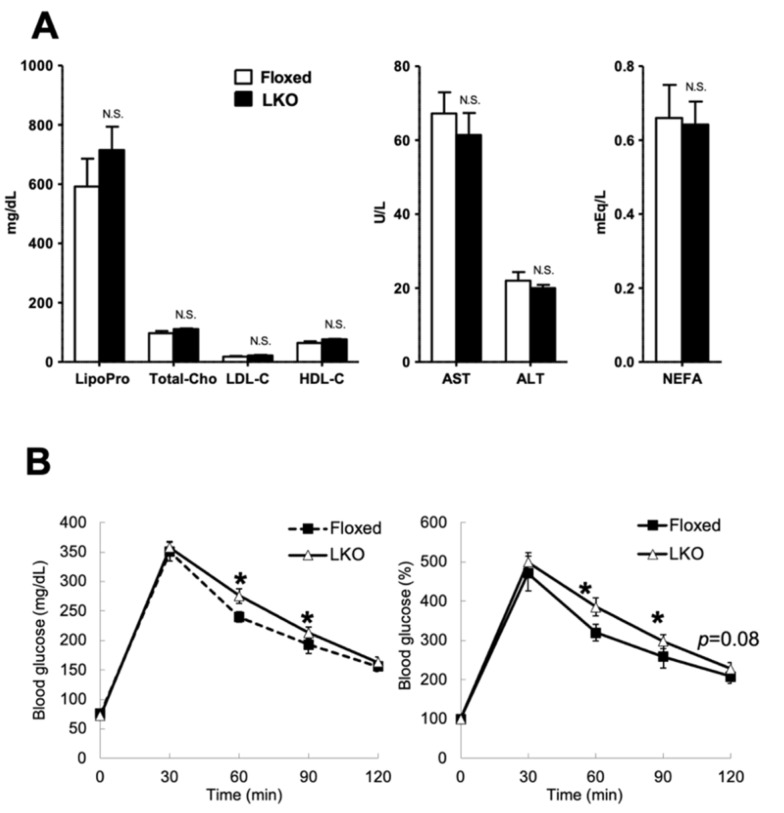
Measurement of serum biochemical markers. (**A**) The values of the serum biochemical test of lipoprotein (Lipopro), total cholesterol (Total-Cho), LDL cholesterol (LDL-C), HDL cholesterol (HDL-C), aspartate transaminase (AST), alanine transaminase (ALT), and non-esterified fatty acid (NEFA) (*n* = 6 each). (**B**) Blood glucose level in Floxed (black square node) and LKO (white triangle node) mice after glucose administration (Floxed: *n* = 4, LKO: *n* = 6). Left graph shows the concentration of blood glucose, and right graph shows the rate of increase of blood glucose. Student’s *t*-test, * *p* < 0.05. N.S.: not significant.

**Figure 4 nutrients-13-03468-f004:**
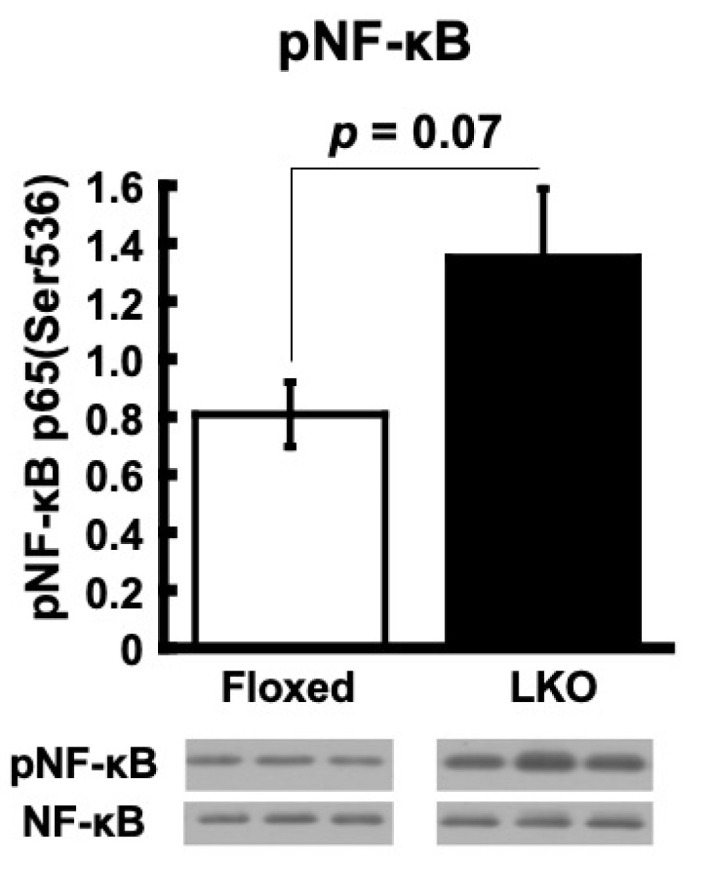
*Phgdh* deletion in hepatocytes induces inflammation and stress response in the liver. Protein levels of phosphorylated NK-κB in the liver of Floxed and LKO mice at 30 weeks (*n*= 6 each). Comparable staining of NK-κB was used to verify equivalent protein loading.

**Figure 5 nutrients-13-03468-f005:**
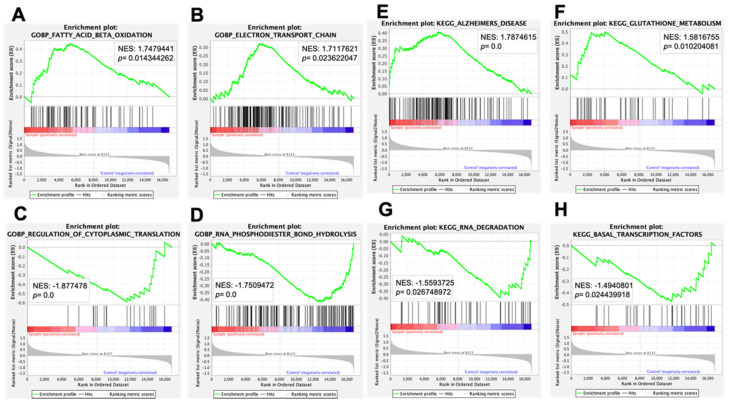
Identification of positively and negatively correlated gene sets in the liver of LKO mice in GO terms and KEGG pathways by gene set enrichment analysis (GSEA). The enrichment plots of GSEA showed positively correlated gene sets in fatty acid beta oxidation (**A**), electron transport chain (**B**), negatively correlated gene sets in regulation of cytoplasmic translation (**C**), and RNA phosphodiester bond hydrolysis (**D**) of GO terms. The enrichment plots of GSEA showed positively correlated gene sets in Alzheimer’s disease (**E**) and glutathione metabolism (**F**) and negatively correlated gene sets in RNA degradation (**G**) and basal transcription factors (**H**) of KEGG pathways. Nominal enrichment scores (NESs) and *p*-values are indicated in each enrichment plot.

**Figure 6 nutrients-13-03468-f006:**
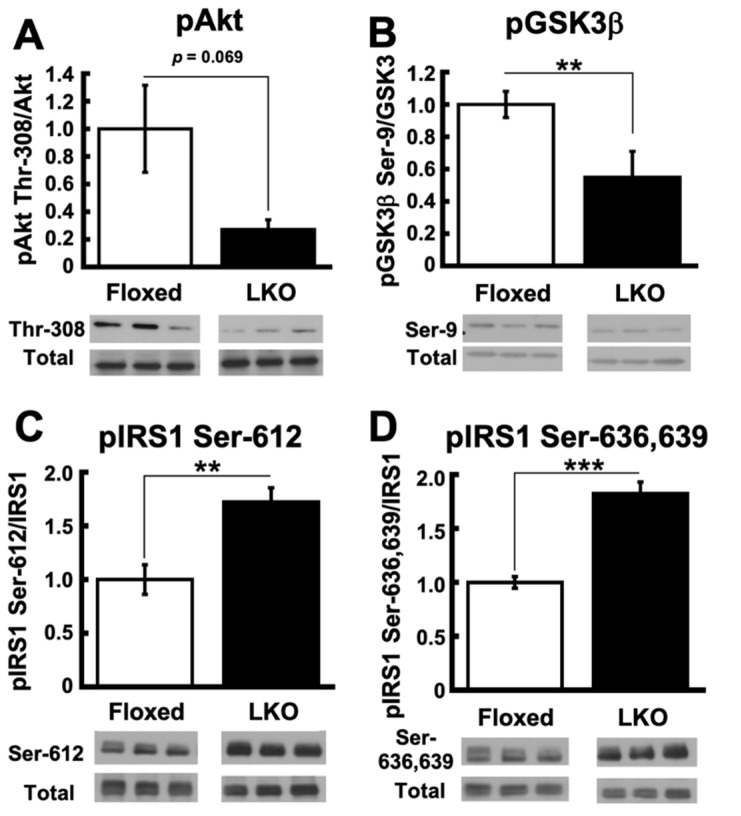
*Phgdh* deletion in hepatocytes impairs insulin signaling in the liver. (**A**) Protein level of phosphorylated Akt in the liver of Floxed and LKO mice at 30 weeks. Comparable staining of Akt was used to verify equivalent protein loading. (**B**) Protein level of phosphorylated GSK3β in the liver of Floxed and LKO mice at 30 weeks. Comparable staining of GSK3β was used to verify equivalent protein loading. (**C**,**D**) Protein level of phosphorylated IRS-1 on Ser612 (C) and Ser636/639 (**D**) residue in the liver of Floxed and LKO mice at 30 weeks. Comparable staining of IRS-1 was used to verify equivalent protein loading. *n* = 6 each. Student’s *t*-test, ** *p* < 0.005, *** *p* < 0.0005.

**Figure 7 nutrients-13-03468-f007:**
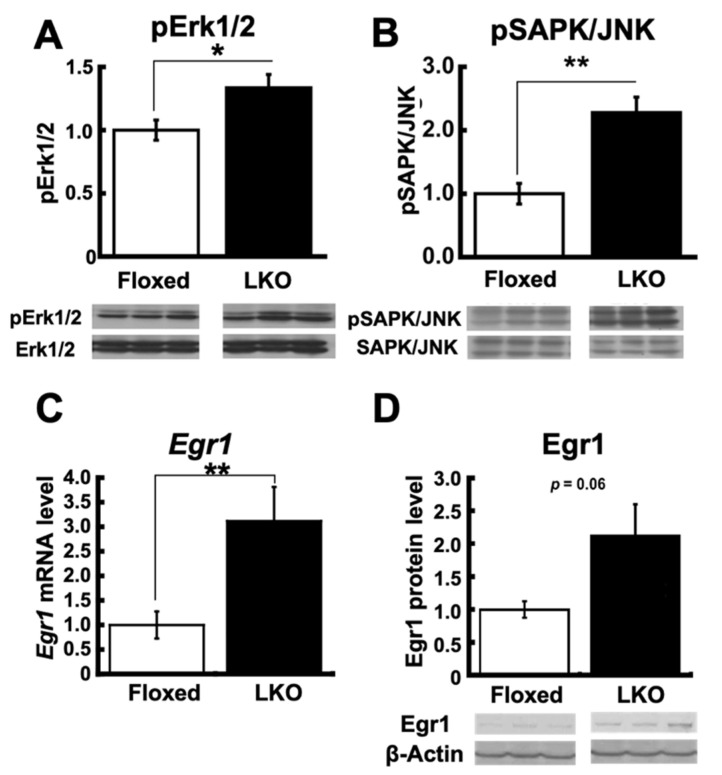
*Phgdh* deletion in hepatocytes alters signaling pathway upstream of IRS1 in liver. (**A**) Protein level of phosphorylated Erk1/2 in the liver of Floxed and LKO mice at 30 weeks. Comparable staining of Erk1/2 was used to verify equivalent protein loading. (**B**) Protein level of phosphorylated SAPK/JNK in the liver of Floxed and LKO mice at 30 weeks. Comparable staining of SAPK/JNK was used to verify equivalent protein loading. (**C**,**D**) mRNA (**C**) and protein (**D**) level of Egr-1 in the liver of Floxed and LKO mice at 30 weeks. *N* = 6 each. Student’s *t*-test, * *p* < 0.05, ** *p* < 0.005.

**Figure 8 nutrients-13-03468-f008:**
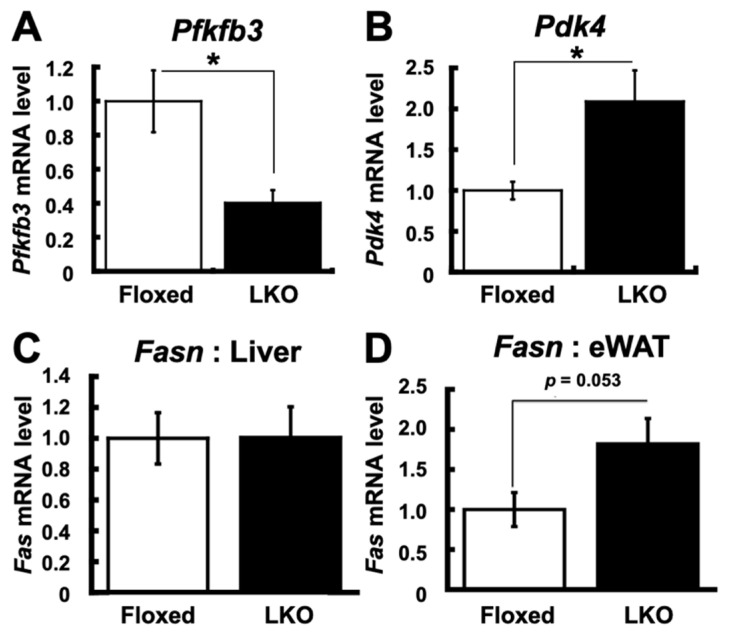
*Phgdh* deletion in hepatocytes induces the reduction of glycolysis in the liver. (**A**,**B**) mRNA levels of *Pfkfb3* (**A**) and *Pdk4* (**B**) in the liver of Floxed and LKO mice at 30 weeks. (**C**,**D**) mRNA levels of *Fasn* in liver(**C**) and eWAT (**D**) in the liver of Floxed and LKO mice at 30 weeks. *n* = 6 each. Student’s *t*-test, * *p* < 0.05.

**Figure 9 nutrients-13-03468-f009:**
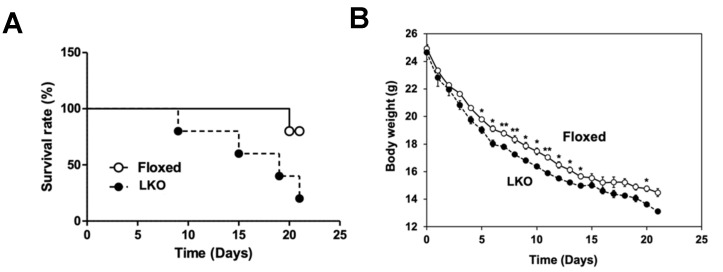
Effect of body weight and survival rate by feeding LKO mice a protein-free diet. (**A**) The transition of body weight in Floxed mice (white node) and LKO mice (black node) after feeding protein-free diet. (**B**) The transition of survival rate in Floxed and LKO mice during feeding protein-free diet. *n* = 5 each. Student’s *t*-test, * *p* < 0.05, ** *p* < 0.005.

**Figure 10 nutrients-13-03468-f010:**
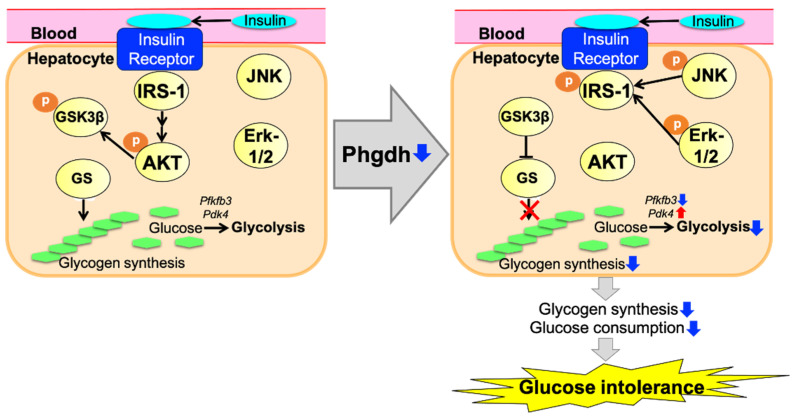
A summary of the molecular mechanisms caused by *Phgdh* deletion in hepatocytes, as inferred by this study. *Phgdh* deletion in hepatocytes impairs insulin signaling via IRS-1 phosphorylation. Akt inactivation by impaired insulin signaling induced the suppression of gluconeogenesis. Enhanced glycolysis resulted in mild obesity and glucose intolerance.

**Table 1 nutrients-13-03468-t001:** Evaluation of amino acid metabolism in the liver and kidney of LKO mice.

**(A)**				
**Amino Acid Concentration in Liver**				
	**Concentration** **(nmol/g Weight Tissue)**			
**Amino Acid**	**Floxed Group**	**LKO Group**	**Ratio** **(%: LKO/Floxed)**	***p*-Value**
L-Asp	79.17 ± 11.29	105.21 ± 14.55	132.9	N.S.
L-G1u	713.44 ± 97.8	731.2 ± 107.57	102.5	N.S.
L-Ser	323.55 ± 39.72	343.51 ± 37.4	106.2	N.S.
L-Gln	2946.45 ±308.2	3167.27 ± 174.77	107.5	N.S.
L-His	364.72 ± 6.91	414.93 ± 28.46	113.8	N.S.
L-Thr	234.27 ± 17.43	224.61 ± 15.36	95.9	N.S.
Gly	1715.62 ± 49.12	1717.33 ± 116	100.1	N.S.
L-Arg	26.97 ± 14.24	29.42 ± 18.42	109.1	N.S.
Tau	9200.13 ± 823.4	7966.99 ± 1591.07	86.6	N.S.
GABA	130.97 ± 2.42	135.75 ± 3.4	103.7	N.S.
L-Ala	2618.32 ± 210.7	2746.58 ± 140.02	104.9	N.S.
L-Tyr	248.25 ± 21.96	238.91 ± 13.9	96.2	N.S.
L-Val	399.78 ± 100.8	455.24 ± 169.22	113.9	N.S.
L-Met	68.41 ± 8.43	64.92 ± 6.41	94.9	N.S.
L-Phe	249.84 ± 16	237.87 ± 12.7	95.2	N.S.
L-Ile	99.47 ± 8.42	104.51 ± 8.33	105.1	N.S.
L-Leu	496.8 ± 36.46	456.04 ± 21.95	91.8	N.S.
**(B)**				
**Amino Acid Concentration in Kidney**				
	**Concentration** **(nmol/g Weight Tissue)**			
**Amino Acid**	**Floxed Group**	**LKO Group**	**Ratio** **(%: LKO/Floxed)**	***p*-Value**
L-Asp	1464.3 ± 162.39	1439.86 ± 89.12	98.3	0.02
L-G1u	4171.94 ± 321.28	3847.12 ± 214.57	92.2	N.S.
L-Ser	502.84 ± 41.5	702.08 ± 41.04	139.6	0.007
L-Gln	847.54 ± 39.02	937.62 ± 40.59	110.6	N.S.
L-His	94.1 ± 10.05	127.82 ± 9.09	135.8	0.009
L-Thr	300.35 ± 25.49	354.49 ± 19.05	118.0	N.S.
Gly	3641.13 ± 182.57	3830.17 ± 273.83	105.2	N.S.
L-Arg	178.67 ± 22.76	250.48 ± 17.65	140.2	0.03
Tau	5757.92 ± 238.44	5924.89 ± 366.5	102.9	N.S.
GABA	81.84 ± 1.09	95.92 ±6.36	117.2	0.054
L-Ala	983.93 ± 66.02	1126.23 ± 53.74	114.5	0.07
L-Tyr	350.66 ± 27.68	444.34 ± 27.39	126.7	0.04
L-Val	330.69 ± 91.3	406.58 ± 80.47	122.9	N.S.
L-Met	75.69 ± 8.7	106.19 ± 8.98	140.3	0.03
L-Phe	187.23 ± 12.96	248.64 ± 15.72	132.8	0.01
L-Ile	104.43 ± 5.86	135.03 ± 9.53	129.3	0.02
L-Leu	430.37 ± 25.3	562.25 ± 31.6	130.6	0.009

(**A**,**B**) Amino acid concentrations in Floxed and LKO mice in the liver (**A**) and kidney (**B**) (*n* = 6 each). The ratio shows each amino acid concentration level in LKO mice compared to that in Floxed mice. Statistical analysis was performed using the Student’s *t*-test. N.S. indicates not statistically significant. l-Asp: L-Asparatic acid, l-Glu: Glutamic acid, l-Ser: Serine, l-Gln, Glutamine, l-His: Histidine, l-Thr: Threonine, Gly: Glycine, l-Arg: l-Arginine, Tau: Taurine, GABA: γ(gamma)-aminobutyric acid, l-Ala: l-alanine, l-Tyr: l-tyrosine, l-Val: l-valine, l-Met: l-methionine, l-Phe: l-phenylalanine, l-Ile: l-isoleucine, L-Leu: l-leucine.

**Table 2 nutrients-13-03468-t002:** Detection of altered signaling pathway in the liver of LKO mice.

**(A)**	
**Enriched KEGG Pathway in Up-Regulated Genes**	
**Term**	***p*-Value**
mmu04740:O1factory transduction	2.90 × 10^−18^
mmu03320:PPAR signaling pathway	0.02511031
mmu04360:Axon guidance	0.04367046
**(B)**	
**Enriched KEGG Pathway in Down-Regulated Genes**	
**Term**	***p*-Value**
mmu05211:Renal cell carcinoma	0.00163639
mmu03015:mRNA surveillance pathway	0.00207197
mmu05220:Chronic myeloid leukemia	0.01099769
mmu04510:Focal adhesion	0.01549065
mmu03018:RNA degradation	0.01850316
mmu04630:Jak-STAT signaling pathway	0.01888063
mmu05221:Acute myeloid leukemia	0.02040881
mmu04151:PI3K-Akt signaling pathway	0.03143164
mmu04152:AMPK signaling pathway	0.03272039
mmu05212:Pancreatic cancer	0.03310616
mmu05200:Pathways in cancer	0.03348629
mmuO4713:Circadian entrainment	0.03641698
mmu04015:Rapl signaling pathway	0.04736727

KEGG pathway enrichment analysis of differentially expressed genes (DEGs) in LKO mice whose expression was upregulated (**A**) or downregulated (**B**) compared to Floxed mice.

**Table 3 nutrients-13-03468-t003:** Detection of altered pathway, toxicity, and disease/disorder by IPA in the liver of LKO mice.

**(A)**		
**Top Canonical Pathway**		
**Name**	***p*-Value**	**Overlap**
IL-12 Signaling and Production in Macrophages	5.96 × 10^−4^	9.8% (11/112)
Ephrin Receptor Signaling	6.14 × 10^−4^	8.3% (14/168)
UVA-Induced MAPK Signaling	9.92 × 10^−4^	10.7% (9/84)
FLT3 Signaling in Hematopoietic Progenitor Cells	1.23 × 10^−3^	11.4% (8/70)
Calcium Signaling	1.27 × 10^−3^	8.1% (13/161)
**(B)**		
**Diseases and Disorders**		
**Name**	***p*-Value**	**# Molecules**
Cancer	0.0304–3.00 × 10^−6^	328
Hematological Disease	0.0304–3.00 × 10^−6^	52
Immunological Disease	0.0304–3.00 × 10^−6^	43
Organismal Injury and Abnormalities	0.0304–3.00 × 10^−6^	338
Tumor Morphology	0.0304–3.00 × 10^−6^	11
**(C)**		
**Hepatotoxicity**		
**Name**	***p*-Value**	**# Molecules**
Liver Regeneration	0.440–0.0228	3
Liver Edema	0.0304	1
Liver Fibrosis	0.306–0.0304	7
Liver Necrosis/Cell Death	0.247–0.0304	11
Hepatocellular Carcinoma	1.00–0.0352	12

IPA detected the top canonical pathway (**A**), disease and disorder (**B**), and hepatotoxicity (**C**) following the DEGs in the livers of LKO mice.

**Table 4 nutrients-13-03468-t004:** Top 20 GO terms and KEGG pathways of differentially expressed genes in the liver of LKO mice. Top 20 GO terms of upregulated (**A**) and downregulated (**B**) genes in the liver of LKO mice compared with Floxed mice. Top 20 KEGG pathways of upregulated genes (**C**) and downregulated genes (**D**) in liver of LKO mice compared with Floxed mice.

**(A)**		
**Name**	**NES**	**NOM *p*-Value**
GOBP_NEGATIVE_REGULATION_OF_NUCLEOCYTOPLASMIC_TRANSPORT	1.7842134	0.00613497
GOBP_BRANCHED_CHAIN_AMINO_ACID_METABOLIC_PROCESS	1.7758015	0.00203666
GOBP_FATTY_ACID_BETA_OXIDATION	1.7479441	0.01434426
GOBP_REGULATION_OF_CAMP_DEPENDENT_PROTEIN_KINASE_ ACTIVITY	1.732202	0
GOBP_ELECTRON_TRANSPORT_CHAIN	1.7117621	0.02362205
GOBP_CELLULAR_METABOLIC_COMPOUND_SALVAGE	1.7083353	0
GOBP_ATP_SYNTHESIS_COUPLED_ELECTRON_TRANSPORT	1.707343	0.04918033
GOBP_NOTOCHORD_DEVELOPMENT	1.6475885	0.00393701
GOBP_COCHLEA_DEVELOPMENT	1.646138	0
GOBP_SECRETION_BY_TISSUE	1.6355267	0.00587084
GOBP_PYRIMIDINE_NUCLEOSIDE_TRIPHOSPHATE_METABOLIC_PROCESS	1.6341659	0.01185771
GOBP_PYRIMIDINE_RIBONUCLEOSIDE_TRIPHOSPHATE_METABOLIC_PROCESS	1.6233511	0.02564103
GOBP.METANEPHRIC_N EPHRON_MORPHOGEN ESIS	1.6204721	0.01996008
GOBP_REGULATION_OF_CARDIAC_CONDUCTION	1.6196082	0.01859504
GOBP_RESPIRATORY_ELECTRON_TRANSPORT_CHAIN	1.6190714	0.076
GOBP_SPERM_EGG_RECOGNITION	1.6184356	0.00592885
GOBP_DNA_UNWINDING_INVOLVED_IN_DNA_REPLICATION	1.618192	0.01030928
GOBP_METANEPHROS_MORPHOGENESIS	1.6089716	0.01629328
GOBP_MONOVALENT_INORGANIC_ANION_HOMEOSTASIS	1.6088727	0.00804829
GOBP_PYRIMIDINE_NUCLEOSIDE_TRIPHOSPHATE_BIOSYNTHETIC_PROCESS	1.6057541	0.01207244
**(B)**		
**Name**	**NES**	**NOM *p*-Value**
GOBP_REGULATION_OF_CYTOPLASMIC_TRANSLATION	−1.877478	0
GOBP_RNA_PHOSPHODIESTER_BOND_HYDROLYSIS_EXONUCLEOLYTIC	−1.8400294	0.002
GOBP_MATURATION_OF_5_8S_RRNA_FROM_TRICISTRONIC_RRNA_TRANSCRIPT_SSU_RRNA_5_8S_RRNA_LSU_RRNA	−1.8186126	0.00412371
GOBP_NEGATIVE_REGULATION_OF_PROTEIN_TYROSINE_KINASE_ACTIVITY	−1.8014549	0
GOBP_RNA_PHOSPHODIESTER_BOND_HYDROLYSIS	−1.7509472	0
GOBP_CLEAVAGE_INVOLVED_IN_RRNA_PROCESSING	−1.7341155	0.01649485
GOBP_POSITIVE_REGULATION_OF_VIRAL_TRANSCRIPTION	−1.7214938	0.0260521
GOBP_REGULATION_OF_MACROPHAGE_CHEMOTAXIS	−1.7151726	0
GOBP_PEPTIDYL_LYSINE_ ACETYLATION	−1.7136337	0
GOBP_MRNA_CLEAVAGE	−1.6955862	0.01
GOBP_POSITIVE_REGULATION_OF_HISTONE_DEACETYLATION	−1.684807	0.00984252
GOBP_NUCLEAR_TRANSCRIBED_MRNA_CATABOLIC_PROCESS_EXONUCLEOLYTIC	−1.680216	0.006
GOBP_VIRAL_GENE_EXPRESSION	−1.6742575	0.02615694
GOBP_MATURATION_OF_5_8S_RRNA	−1.6727061	0.02340426
GOBP_TRANSCRIPTION_PREINITIATION_COMPLEX_ASSEMBLY	−1.6675799	0.01757813
GOBP_PROTEIN_LIPID_COMPLEX_ASSEMBLY	−1.6524748	0.01335878
GOBP_NUCLEAR_ENVELOPE_REASSEMBLY	−1.6334432	0.03012048
GOBP_PROTEIN_ACETYLATION	−1.6297097	0
GOBP_PEPTIDYL_ASPARAGINE_MODIFICATION	−1.6176745	0.02985075
GOBP_TRANSEPITHELIAL_TRANSPORT	−1.6128986	0.00395257
**(C)**		
**Name**	**NES**	**NOM *p*-Value**
KEGG_ALZHEIMERS_DISEASE	1.7874615	0
KEGG_CARDIAC_MUSCLE_CONTRACTION	1.6276087	0.00796813
KEGG_VALINE_LEUCINE_AND_ISOLEUCINE_DEGRADATION	1.6064283	0.02484472
KEGG_GLUTATHIONE_METABOLISM	1.5816755	0.01020408
KEGG_PARKINSONS_DISEASE	1.5710168	0.10224949
KEGG_HUNTINGTONS_DISEASE	1.4995617	0.03952569
KEGG_FATTY_ACID_METABOLISM	1.4914919	0.04208417
KEGG_GLYCEROLIPID_METABOLISM	1.4762139	0.05633803
KEGG_PEROXISOME	1.4749482	0.125
KEGG_OXIDATIVE_PHOSPHORYLATION	1.4733046	0.14229248
KEGG_ARACHIDONIC_ACID_METABOLISM	1.4536077	0.00626305
KEGG_PROPANOATE_METABOLISM	1.446644	0.11332008
KEGG_OLFACTORY_TRANSDUCTION	1.4194456	0.02385686
KEGG_PPAR_SIGNALING_PATHWAY	1.4145677	0.06412826
KEGG_TRYPTOPHAN_METABOLISM	1.409311	0.12352941
KEGG_REGULATION_OF_AUTOPHAGY	1.4061221	0.07272727
KEGG_GLYCOSAMINOGLYCAN_BIOSYNTHESIS_HEPARAN_SULFATE	1.383326	0.05371901
KEGG_DNA_REPLICATION	1.3792615	0.17693837
KEGG_CALCIUM_SIGNALING_PATHWAY	1.362877	0.03193613
KEGG_GNRH_SIGNALING_PATHWAY	1.3259736	0.07628866
**(D)**		
**Name**	**NES**	**NOM *p*-Value**
KEGG_DORSO_VENTRAL_AXIS_FORMATION	−1.6805534	0.00199601
KEGG_RNA_DEGRADATION	−1.5593725	0.02674897
KEGG_NON_SMALL_CELL_LUNG_CANCER	−1.5283813	0.03092784
KEGG_N_GLYCAN_BIOSYNTHESIS	−1.5252374	0.06681035
KEGG_BASAL_TRANSCRIPTION_FACTORS	−1.4940801	0.02443992
KEGG_GLYCOSYLPHOSPHATIDYLINOSITOL_GPI_ANCHOR_BIOSYNTHESIS	−1.4563912	0.05285412
KEGG_PANCREATIC_CANCER	−1.4190394	0.05020081
KEGG_RENAL_CELL_CARCINOMA	−1.3920702	0.0831643
KEGG_ADHERENSJUNCTION	−1.3889191	0.09543569
KEGG_ENDOMETRIAL_CANCER	−1.3684485	0.05702648
KEGG_PROSTATE_CANCER	−1.3127599	0.10655738
KEGG_SMALL_CELL_LUNG_CANCER	−1.2742031	0.12576064
KEGG_TGF_BETA_SIGNALING_PATHWAY	−1.2639772	0.16232465
KEGG_SPLICEOSOME	−1.2469473	0.18526316
KEGG_PROTEIN_EXPORT	−1.2308676	0.32635984
KEGG_CHRONIC_MYELOID_LEUKEMIA	−1.22399	0.1523046
KEGG_MTOR_SIGNALING_PATHWAY	−1.2147567	0.17979798
KEGG_STEROID_HORMONE_BIOSYNTHESIS	−1.2091544	0.13541667
KEGG_RNA_POLYMERASE	−1.1975825	0.27021277
KEGG_PORPHYRIN_AND_CHLOROPHYLL_METABOLISM	−1.196312	0.21991701

## Data Availability

Microarray data have been deposited in NCBI’s Gene Expression Omnibus and are accessible through GEO series accession number GSE179912.

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
