# Peer review of "Hepatocyte-Specific *Phgdh*-Deficient Mice Culminate in Mild Obesity, Insulin Resistance, and Enhanced Vulnerability to Protein Starvation"

_nutrients, 2021, doi:10.3390/nu13103468_

Round 1

Reviewer 1 Report

The paper demonstrated that Phgdh-dependent de novo Ser synthesis in liver hepatocytes contributes to the maintenance of systemic glucose tolerance, suppression of inflammatory response, and resistance to protein starvation. The paper is well written in general, but I would suggest that the authors ellaborate a little bit more on the potential clinical significance of their findings. 

Author Response

We would like to appreciate reviewer #1 for his/her very positive comments. In accordance with the comments, we have added the following paragraphs to the end of the Discussion section regarding the potential clinical contribution of the present observations.

Discussion (Lines 620-628)

In conclusion, this study reveals that de novo serine synthesis initiated by Phgdh in liver hepatocytes contributes to maintaining the insulin/IGF signaling pathway in the liver, systemic glucose tolerance, and resilience to protein deprivation. Since an impairment in systemic glucose tolerance serve as a high risk for developing type 2 diabetes mellitus in humans, it is anticipated that elucidation of the pathobiological relationship between decrease in serine synthesis via the phosphorylation pathway, diminishment of the insulin/IGF signaling pathway, and impaired glucose tolerance in the human liver will provide new mechanisms related to impaired glucose tolerance and useful opportunities for its diagnosis and prevention.

Reviewer 2 Report

I think the authors have done extensive research and interesting work. I just want to recommend a few small modifications.

Line 61: I recommend to add “Sydrome” after “Neu-Laxova” (“Neu-Laxova Syndrome”).

Materials and methods. Histological evaluation: How many mice were used for this evaluation?

Lines 280-282: The authors write “The blood glucose levels in the LKO mice were significantly higher than those in the Floxed mice after oral glucose administration”, but in “Materials and methods. Glucose tolerance test” they describe that the glucose tolerance test was performed by intraperitoneally injecting glucose. Could they explain this?

Figure 4A: It is impossible to distinguish anything in this scheme. I recommend that the authors add it as supplementary material so that it can be a larger image that is better visualized.

Table 4: The first time the authors refer to table 4 in the text is on line 361 of page 11, however, it does not appear until page 17. I suggest that this table appears after table 3.

Figure 10: This figure appears before being referred to in the text. I suggest that this figure appears after line 498, maybe after line 528.

Figure S1: I think the best way to see the difference between the adipose tissue of Floxed and LKO mice is to place the photographs of both side by side, not on 2 different pages.

Author Response

We would like to appreciate reviewer #2 for his/her valuable comments.

> I think the authors have done extensive research and interesting work. I just want to recommend a few > small modifications.

> Line 61: I recommend to add “Sydrome” after “Neu-Laxova” (“Neu-Laxova Syndrome”).

As you suggested, we changed “Neu-Laxova patient” to “Neu-Laxova Syndrome patient” in the second paragraph of the Introduction section.

> Materials and methods. Histological evaluation: How many mice were used for this evaluation?

We have used 5 Floxed mice and 5 LKO mice for histological evaluation. We have added the number of mice for histological evaluation in the caption of Figure 2 and the Material and Methods section.

Materials and Methods (Lines 222-224)

Mice (male, 28–32 weeks old; n = 5 in each genotype) were anesthetized with isoflurane and perfused with 4% paraformaldehyde in 0.1 M sodium phosphate buffer (pH 7.2) after removing blood with 0.1 M sodium phosphate buffer (pH 7.2).

Figure legends (Lines295-296)

(C, D) Histological evaluation of the liver in Floxed (C) and LKO (D) mice at 28 weeks (n=5 each). Representative images are shown.

> Lines 280-282: The authors write “The blood glucose levels in the LKO mice were significantly higher > than those in the Floxed mice after oral glucose administration”, but in “Materials and methods.    > Glucose tolerance test” they describe that the glucose tolerance test was performed                > by intraperitoneally injecting glucose. Could they explain this?

As reviewer #2 pointed out, the sentence in Lines 280-282 was incorrect. We have performed glucose tolerance test by injecting glucose intraperitoneally. We have corrected “The blood glucose levels in the LKO mice were significantly higher than those in the Floxed mice after oral glucose administration” to “The blood glucose levels in the LKO mice were significantly higher than those in the Floxed mice after intraperitoneal glucose injection.” (lines 284-286)

> Figure 4A: It is impossible to distinguish anything in this scheme. I recommend that the authors add it > as supplementary material so that it can be a larger image that is better visualized.

As your suggested, we have moved Figure 4A into supplementary material as supplementary Figure 3.

> Table 4: The first time the authors refer to table 4 in the text is on line 361 of page 11, however, it does > not appear until page 17. I suggest that this table appears after table 3.

As your suggested, we changed the Table 4 layout after Table 3.

> Figure 10: This figure appears before being referred to in the text. I suggest that this figure appears   > after line 498, maybe after line 528.

As your suggested, we changed the Figure 10 layout into the Discussion section.

> Figure S1: I think the best way to see the difference between the adipose tissue of Floxed and LKO   > mice is to place the photographs of both side by side, not on 2 different pages.

As you suggested, we changed the figure layout in supplementary Figure 1 to place the photographs of Floxed and LKO mice side by side.
